# Clinicopathological and molecular subtypes of breast cancer in the Eastern Cape, South Africa: A two-year retrospective study

Hayden Gerald Kretzmann[1,2]*, Oladele Vincent Adeniyi[2,3]

1 Division of General Surgery, Department of Surgery, Frere Hospital, East London, South Africa,
2 Faculty of Medicine & Health Sciences, Walter Sisulu University, East London, South Africa,
3 Department of Family Medicine, Cecilia Makiwane Hospital, East London, South Africa.

* hgkretzmann@gmail.com

## Abstract

### Background

Breast cancer (BC) is the most common cancer in women worldwide and the most frequent cause of cancer death in women in low- and middle-income countries (LMIC). The incidence of BC in Africa is on the rise, expected to double by 2050, primarily owing to late presentation and weak health infrastructure in sub-Saharan Africa (SSA). This study addresses the lack of recent data on BC cases in the Eastern Cape Province of South Africa.

### Objective

The objectives of this study were to describe the clinicopathological characteristics and molecular subtypes of BC and, in addition, to examine the association between the clinicopathological characteristics and the molecular subtypes of BC in a single tertiary hospital in the Eastern Cape Province of South Africa.

### Methods

A two-year (2022–2023) retrospective cross-sectional clinical record review study was conducted on patients treated for invasive BC at a tertiary hospital in the Eastern Cape Province, South Africa. The demographic, clinical and pathological characteristics and molecular subtypes were reported. Associations were investigated between the BC molecular subtypes identified and the clinicopathological characteristics of the patients.

### Results

A total of 282 patients met the study's inclusion criteria. Most patients were female (98.6%) and African (88.1%). The mean age of the patients was 58.7 years, with BC

**Data availability statement:** All relevant data are within the manuscript and its Supporting Information files.

**Funding:** The author(s) received no specific funding for this work.;

**Competing interests:** The authors have declared that no competing interests exist.

most prevalent in the age group >70 (25.2%) and postmenopausal (77.4%). Breast lump was the most common presenting complaint (98.6%), with 61% of patients presenting three months after noticing the anomaly. The most common tumour size (59.4%) was > 5 cm (mean = 6.37 ± 3.6), with the most common clinical T stage being T4 (50.4%). Lymph node involvement was seen in 50.4% of cases. Patients mostly presented in Stages III and IV of the disease (60.1%). Invasive ductal carcinoma not otherwise specified (NOS) was the most common histopathological subtype (86.2%). Grade 2 (56.2%) and Grade 3 (29.5%) BC accounted for the majority of cases. Luminal B was found in 47.4% of cases, Luminal A in 28.5%, triple negative breast cancer (TNBC) in 18.6% and human epidermal growth factor receptor 2 (HER2) enriched in 5.5% of cases, respectively.

## Conclusion

In our setting, most patients consulted at a late stage of the disease with a large tumour size, positive lymph node status and a high histological grade. Luminal B tumours are the most common molecular subtype. These results indicate the need for more intensive breast cancer awareness campaigns, early detection, and timely referral and treatment.

## Introduction

Breast cancer (BC) is now the most commonly diagnosed cancer among women globally and the leading cause of cancer deaths among women in low- and middle-income countries (LMIC) in sub-Saharan Africa (SSA) [1–6]. In 2022, GLOBOCAN reported that there were 2.3 million BC cases globally, contributing to 11.6% of all newly diagnosed cancers. BC was the fourth leading cause of cancer mortality worldwide, causing 666 000 deaths in 2022 (6.9% of all cancer deaths) [1].

BC incidence rates vary significantly between regions, and high-income countries (HIC) have a four times higher incidence than LMIC such as South Africa (SA), (54.1 vs 30.8 per 100 000, respectively). However, HIC's also have significantly lower mortality rates (11.3 vs 15.3 per 100 000) [1,7]. Globally, there is an increasing incidence of BC owing to changes in the prevalence of risk factors – population growth, improved life expectancy, obesity and increased use of hormonal contraceptives [8] – and increased detection by mammographic screening and improved treatment [7]. In SSA, the BC incidence is projected to double, reaching 3.2 million cases by 2050 [9–11]. The at-risk population is steadily increasing in SSA countries, with countries like Nigeria seeing annual increases of 2.5%. This is in contrast with high-income countries (HIC) such as France, where the at-risk population has plateaued [3].

BC in these regions is complex and heterogeneous, exhibiting poorer prognostic features more frequently than in HIC's, including younger age at diagnosis, larger tumour sizes, higher tumour grades, more positive lymph nodes status, higher prevalence of triple-negative (TNBC) and human epidermal growth factor receptor

2 (HER2) positive molecular subtypes, and later clinical stages at diagnosis [12–14]. Although BC survival rates have improved globally [6], they remain low in SSA, with the African Breast Cancer – Disparities in Outcome study revealing a three-year overall survival rate of 50% [15]. In South Africa, a significant proportion of patients present at advanced stages, affected by limited access to screening and healthcare services [16]. The Eastern Cape Province lacks comprehensive epidemiological data of the BC patients, highlighting a knowledge gap and the need for further research to improve early detection and management efforts in the region. This work aims to document the clinicopathological characteristics and molecular subtypes of invasive BC cases in the Eastern Cape Province. In addition to examine the association between the clinicopathological characteristics and BC molecular subtypes.

## Methods

### Design, setting and population

In this retrospective cross-sectional study, we reviewed the records of all 282 patients treated for invasive breast cancer (BC) at the radiation oncology combined BC unit of Frere Hospital, East London, over a two-year period from 1 January 2022–31 December 2023. The radiation oncology combined BC unit at Frere Hospital provides public tertiary services to the central region of the Eastern Cape Province, South Africa. This region encompasses four districts: Buffalo City Metropolitan Municipality, Amathole Municipality, Chris Hani Municipality, and Joe Gqabi Municipality. It has a combined population of 3 068 291 as of 2022 [17].

### Inclusion and exclusion criteria

The patients included in the study comprised all histologically confirmed primary invasive BC cases treated at the radiation oncology combined BC unit during the study period. Patients were excluded if they had premalignant disease or benign disease, or metastasis to the breast.

### Data collection

Data was collected from patient medical records and pathological reports stored in the oncology database, over the time period 1 November 2024–15 December 2024, and transferred to a Microsoft Excel spreadsheet. Data was anonymised from patient folders to the Microsoft Excel spreadsheet as folder numbers were used to preserve patient anonymity and the data was store on a password protected computer only accessible by the author and supervisor to maintain data protection. Relevant items on demographic characteristics (age, gender, race and type of residence), medical history (menopausal status, family history of BC and family history of other cancers) clinical characteristics (presenting complaint, duration of symptoms, breast side, affected breast quadrant, tumour size, lymph node status), complete histopathological reports and treatment were received. All patients had a biopsy to determine the complete immunohistochemical evaluation before initiation of treatment. Tumour–node–metastasis (TNM) staging was extracted from the records in accordance with the American Joint Committee on Cancer Staging Manual 8th edition [18]. The tumour grade was classified according to the Elston-Ellis modification of the Scarff-Bloom and Richardson grading system [19]. Tumour receptor status was documented and subsequently categorised into molecular subtypes as follows: Luminal A (ER + , PR + /PR–, Ki 67 < 20%, HER2–), Luminal B (ER + , PR + /PR–, Ki 67 > 20%, HER2±), human epidermal growth factor receptor 2 (HER2) enriched (ER–, PR–, HER2+) and triple negative breast cancer (TNBC) (ER–, PR–, HER2–) [20].

Ethical clearance (172/2024) was granted by the Walter Sisulu University (WSU) Faculty of Health Sciences Research Ethics and Biosafety Committee. The study protocol followed the Helsinki Declaration and Good Clinical Practice Guidelines. The need for consent was waived by the ethics committee. The anonymized data set used for the analysis is provided as supporting information, S1 Table.

## Statistical analysis

Microsoft Excel was used for data capturing, with statistical analysis conducted using Stata version 18.0. Statistical significance was set at a p-value of <0.05. A 95% confidence interval (CI) was used. Descriptive statistics were used for data analysis. For numerical data, central tendency and variability were described using means and standard deviations if normally distributed, and medians and interquartile ranges (IQRs) if not normally distributed. The Fisher Exact test was used to assess the association of the clinicopathological characteristics with the BC molecular subtypes. Due to the retrospective nature of the study, a sample size of 219 was calculated using the Cochrane formula. As some data sets were expected to be incomplete, the sample size was adjusted by a factor of 30% to account for this (n = 285). The two Luminal B groups (Luminal B HER2- and Luminal B HER2+ groups) were combined to acquire greater numbers to allow the Fisher Exact test to run and obtain a p-value.

## Results

Data were available for 282 patients from 1 January 2022–31 December 2023. Almost all the patients were females (98.6%; n = 278). Given the small proportion of males (n = 4; 1.4%), further analysis was restricted to the female category. The mean age was 58.7 years (standard deviation (SD) ±14.9) and the median age was 59.5 years (interquartile range of 31–94). The youngest patient in this study was 24 years old and the oldest was 99 years old. The peak incidence of breast cancer (BC) occurred in the >70 years age group (25.2%). The most common ethnicity was African (n = 245; 88.1%). Most women were post-menopausal (n = 206; 77.4%) Table 1.

The most common presentation of BC was a breast lump, either alone (98.6%), combined with skin changes (44.7%), or combined with mastalgia (33.7%). Most patients presented after three months' duration (61.0%). The mean tumour size at presentation was 6.37 cm (SD ± 3.63), tumour size being >5 cm in 165 patients (59.4%), with the most common clinical T stage being advanced T4 lesions (50.4%). About half of the women had palpable axillary lymph nodes at diagnosis (n = 140; 50.4%), and of the positive lymph node group, the most common clinical node (N) stage was N1 (n = 73; 26.3%). Most women presented at an advanced stage of the disease (Stages III and IV) (n = 167; 60.1%); n = 134 (48.2%) for clinical Stage III, and n = 33 (11.9%) for clinical Stage IV. At presentation, 33 women had metastasis (11.9%), with the majority presenting at late-stage BC (IIb, III and IV) (n = 212; 76.3%) Table 2.

Invasive ductal carcinoma not otherwise specified (NOS) was the most prevalent histological type of BC in the study sample (n = 238; 86.2%), followed by invasive lobular carcinoma 14 (n = 14; 5.1%) and papillary carcinoma (n = 10; 3.6%) in the female population. Most tumours were graded as histological Grade 2 (n = 128; 56.2%) and Grade 3 (n = 68; 29.8%). In total, 157 patients received surgical treatment (56.5%); a substantial majority received modified radical mastectomies 150 (n = 150; 95.5%) and the remaining seven (4.5%) received breast conserving surgery (BCS) Table 3.

The most common molecular subtype was Luminal B human epidermal growth factor receptor 2 negative (HER2–) (n = 102; 37.2%), followed by Luminal A (n = 78; 28.5%), triple negative breast cancer (TNBC) (n = 51; 18.6%), and Luminal B (HER2+) (n = 28; 10.2%) Table 4.

No significant association could be demonstrated between the demographic characteristics of the patients and the BC molecular subtypes, Table 5.

In the Fisher's exact test, the duration of the presenting complaint showed a significant relationship with the BC molecular subtype; the HER2 enriched and TNBC molecular subtypes tended to present earlier than the Luminal types (p =<0.001). Similarly, the clinical T stage showed a significant relationship with the BC molecular subtypes. All the various molecular subtypes of BC were diagnosed at the most advanced stage (T4) (p = 0.033). The presence of an axillary lump was significantly related to the BC molecular subtype, with an axillary lump present in the majority of HER2 enriched and Luminal B cases (66.7% and 58.5%, respectively), as well in 42.3% of Luminal A and 39.2% of TNBC cases (p = 0.023).

The clinical stage at presentation was found to be significantly related to the BC molecular subtype, with 70.7% of Luminal B cases and 60% of HER2 enriched cases presenting at advanced stages, either Stage III or Stage IV (p = 0.002). Similarly, early and late stages at presentation were significantly related to the BC molecular subtypes. Luminal B and

**Table 1. Demographics characteristics of breast cancer patients.**

| Variables | Frequency (n) | Percentage (%) |
|---|---|---|
| **Gender (N = 282)** | | |
| Male | 4 | 1.4% |
| Female | 278 | 98.6% |
| **Age groups (years) (N = 278)** | | |
| <30 | 3 | 1.1% |
| 30-39 | 25 | 9.0% |
| 40-49 | 53 | 19.1% |
| 50-59 | 65 | 23.4% |
| 60-69 | 62 | 22.3% |
| >70 | 70 | 25.2% |
| **Age (years), mean (std. dev.)** | 58.7 | (±14.9) |
| **Ethnicity (N = 278)** | | |
| African | 245 | 88.1% |
| Coloured | 16 | 5.8% |
| Indian | 1 | 0.4% |
| Caucasian | 15 | 5.4% |
| Asian | 1 | 0.4% |
| ***Menopausal status (N = 266)** | | |
| Pre | 41 | 15.4% |
| Peri | 19 | 7.1% |
| Post | 206 | 77.4% |

*Incomplete data – not available in patient's records

TNBC had the highest percentages of late-stage presentations, at 83.1% and 74.5%, respectively, compared to the Luminal A and HER2 enriched subtypes, which both presented at late stages in 66.7% of cases (p = 0.037) Table 6.

In the Fisher's exact test, the tumour histological type was significantly associated with the BC molecular subtypes. The most common tumour histology was invasive ductal carcinoma not otherwise specified (NOS) across all subtypes; HER2 enriched (100%), Luminal B (92.2%), TNBC (90.2%) and Luminal A (74%). Mucinous (10.4%) and papillary BC (7.8%) were found mostly in Luminal A, and invasive lobular cancer was found mostly in Luminal A BC (7.8%) and Luminal B BC (4.7%) (p = 0.006). Tumour histological grade was also significantly associated with the BC molecular subtype. TNBC and HER2 enriched BC presented with Grade 3 tumours in 72.7% and 61.5% of cases, respectively. Most of the Grade 2 tumours were found in Luminal B cases of BC (70.8%), and the majority of Grade 1 tumours were found in the Luminal A molecular subtype group (40%) (p=<0.001).

Surgical management for BC was significantly related to the molecular subtype; the majority of the Luminal A and TNBC molecular subtypes received mastectomies (68.8% and 62.5%, respectively) (p = 0.037) Table 7.

## Discussion

Breast cancer (BC) is a significant public health issue worldwide, requiring data-driven strategies to mitigate the scourge. However, there is a paucity of epidemiological data on the profile of patients affected by BC in the Eastern Cape Province of South Africa. This study bridges this gap by reporting the characteristics of 282 patients with histologically confirmed BC in the central region of the Eastern Cape Province over a two-year period (2022–2023). The findings yield insights on the current status of BC in the region and open doors for larger studies on the diverse population groups of the country.

**Table 2. Clinical characteristics of patients with breast cancer.**

| Variables | Frequency (n) | Percentage (%) |
|---|---|---|
| **Most common presenting complaint (N = 278) \*** | | |
| Breast lump | 274 | 98.6% |
| Breast lump and skin changes | 122 | 44.7% |
| Breast lump and mastalgia | 90 | 33.7% |
| **Duration of presenting complaint (months) (N = 272)** | | |
| < 3 | 106 | 39.0% |
| 3-6 | 91 | 33.5% |
| 7-9 | 25 | 9.2% |
| 10-12 | 17 | 6.2% |
| >12 | 33 | 12.1% |
| **Tumour size (N = 278)** | | |
| < 2 cm | 23 | 8.3% |
| 2-5 cm | 90 | 32.4% |
| >5 cm | 165 | 59.4% |
| **Mean clinical tumour size (cm)** | 6.37 | 3.633 |
| **Clinical T stage (N = 278)** | | |
| T1 | 21 | 7.6% |
| T2 | 69 | 24.8% |
| T3 | 48 | 17.3% |
| T4 | 140 | 50.4% |
| ➢T4a | 7 | 2.5% |
| ➢T4b | 72 | 25.9% |
| ➢T4c | 38 | 13.7% |
| ➢T4d | 23 | 8.3% |
| **Clinical lymph node status (N = 278)** | | |
| Positive | 140 | 50.4% |
| Negative | 138 | 49.6% |
| **Clinical N stage (N = 278)** | | |
| N0 | 138 | 49.6% |
| N1 | 73 | 26.3% |
| N2 | 51 | 18.3% |
| N3 | 16 | 5.8% |
| **M Stage (N = 277)** | | |
| M0 | 244 | 88.1% |
| M1 | 33 | 11.9% |
| **Clinical TNM stage (N = 278)** | | |
| I | 19 | 6.8% |
| II | 92 | 33.1% |
| III | 134 | 48.2% |
| IV | 33 | 11.9% |
| **Early vs Late stage (N = 278)** | | |
| Early (Stage Ia, Ib, IIa) | 66 | 23.7% |
| Late (Stage IIb, III, IV) | 212 | 76.3% |

T, tumour; N, node; M, metastasis

**Table 3. Pathological characteristics and surgical intervention of breast cancer patients.**

| Variables | Frequency (n) | Percentage (%) |
|---|---|---|
| **Histological type of breast cancer (N = 276)** | | |
| Invasive ductal cancer (NOS) | 238 | 86.2% |
| Invasive lobular cancer | 14 | 5.1% |
| Papillary cancer | 10 | 3.6% |
| Mucinous cancer | 9 | 3.3% |
| Medullary cancer | 1 | 0.4% |
| Other | 4 | 1.4% |
| **Histological grade of tumour (N = 228)** | | |
| Grade 1 | 32 | 14.0 |
| Grade 2 | 128 | 56.2 |
| Grade 3 | 68 | 29.8 |
| **Type of surgery (N = 157)** | | |
| Breast conserving surgery | 7 | 2.6% |
| Mastectomy | 150 | 55.8% |
| Nil | 112 | 41.6% |

NOS, not otherwise specified

**Table 4. Molecular subtype characteristics of breast cancer patients.**

| Variables | Frequency (n) | Percentage (%) |
|---|---|---|
| **Receptor status** | | |
| **ER** | | |
| Yes | 206 | 75.5% |
| No | 67 | 24.5% |
| **PR** | | |
| Yes | 199 | 72.6% |
| No | 75 | 27.4% |
| **HER2** | | |
| Yes | 45 | 16.4% |
| No | 229 | 83.6% |
| **Ki 67** | | |
| <20% | 81 | 29.8% |
| >20% | 191 | 70.2% |
| **Molecular subtype (N = 274)** | | |
| Luminal A | 78 | 28.5% |
| Luminal B (HER2 -) | 102 | 37.2% |
| Luminal B (HER2 +) | 28 | 10.2% |
| HER2 enriched | 15 | 5.5% |
| TNBC | 51 | 18.6% |

ER, oestrogen receptor; PR, progesterone receptor; Ki 67, antigen kiel 67; HER2, human epidermal growth factor receptor 2; TNBC, triple negative breast cancer

**Table 5. Association of patients' breast cancer molecular subtypes with demographic characteristics.**

| Variables | Luminal A | Luminal B | HER2 Enriched | TNBC | Total | P-value* |
|---|---|---|---|---|---|---|
| N | 78 (28.5%) | 130 (47.4%) | 15 (5.5%) | 51 (18.6%) | 274 (100.0%) | |
| **Age group** | | | | | | |
| ≤40 | 6 (7.7%) | 17 (13.1%) | 1 (6.7%) | 4 (7.8%) | 28 (10.2%) | **0.799** |
| 40-59 | 31 (39.7%) | 54 (41.5%) | 7 (46.7%) | 24 (47.1%) | 116 (42.3%) | |
| ≥60 | 41 (52.6%) | 59 (45.6%) | 7 (46.7%) | 23 (45.1%) | 130 (47.45) | |
| **Age groups** | | | | | | |
| <30 | 1 (1.3%) | 0 (0.0%) | 0 (0.0%) | 2 (3.9%) | 3 (1.1%) | **0.849** |
| 30-39 | 5 (6.4%) | 17 (13.1%) | 1 (6.7%) | 2 (3.9%) | 25 (9.1%) | |
| 40-49 | 16 (20.5%) | 22 (16.9%) | 1 (6.7%) | 13 (25.5%) | 52 (19.0%) | |
| 50-59 | 15 (19.2%) | 32 (24.6%) | 6 (40.0%) | 11 (21.6%) | 64 (23.4%) | |
| 60-69 | 14 (17.9%) | 29 (22.3%) | 5 (33.3%) | 13 (25.5%) | 61 (22.3%) | |
| ≥70 | 27 (34.6%) | 30 (23.1%) | 2 (13.3%) | 10 (19.6%) | 69 (25.2%) | |
| **Age** | 60.8 (15.4) | 57.9 (14.8) | 58.0 (12.4) | 57.5 (14.9) | 58.6 (14.8) | **0.524** |
| **Ethnicity** | | | | | | |
| African | 70 (89.2%) | 116 (89.2%) | 13 (86.7%) | 43 (84.3%) | 242 (88.3%) | **0.877** |
| Coloured | 3 (3.8%) | 7 (5.4%) | 1 (6.7%) | 4 (7.8%) | 15 (5.5%) | |
| White | 5 (6.4%) | 6 (4.6%) | 1 (6.7%) | 3 (5.9%) | 15 (5.5%) | |
| Indian/Asian | 0 (0.0%) | 1 (0.8%) | 0 (0.0%) | 1 (2.0%) | 2 (0.7%) | |
| **Menopause** | | | | | | |
| Peri | 3 (4.1%) | 11 (8.7%) | 1 (7.7%) | 4 (8.0%) | 19 (7.2%) | **0.429** |
| Post | 65 (85.1%) | 90 (71.4%) | 10 (76.9%) | 40 (80.0%) | 203 (77.2%) | |
| Pre | 8 (10.8%) | 25 (19.8%) | 2 (15.4%) | 6 (12.0%) | 41 (15.6%) | |

HER2, human epidermal growth factor receptor 2; TNBC, triple negative breast cancer

*p values were calculated using the Fisher exact test

The study revealed a predominantly female BC population (98.6%), with only four males (1.4%) presenting with BC at this hospital during the study period. This finding aligns with the existing literature across Africa, which shows a consistent female preponderance in BC cases [21–24]. A similar prevalence of 1.4% was found among men in a 12-year review of the Kenyan BC registry [22]. Age distribution within this cohort ranged from 24 to 99 years, with a mean age of 58.7 ± 14.9 years. Notably, BC typically occurs 10–15 years earlier in sub-Saharan Africa (SSA) than in high-income countries (HIC), with the median age at diagnosis in SSA being approximately 45 years [10,21,24–26]. The mean age of 58.7 reported in the current study is higher than that documented in other SSA countries. For instance, a study in Kenya found a mean age of 44 years [26], while similar reports from Tanzania and Nigeria indicated mean ages of 44.7 years [10] and 47.5 years [25], respectively. However, the mean age in this study aligns closely with findings from other South African studies. For example, Pietersburg Hospital in Limpopo Province reported a mean age of 55 years [27], and the Charlotte Max-eke Johannesburg Academic Hospital in Gauteng Province recorded a mean age of 57 years [28]. This mean age is still younger than the mean age of 61 years observed in HIC [29]. The peak incidence of BC in this cohort was found among those over 70 years old (25.2%), followed by the 50–59-year age group (23.4%), which is consistent with other findings in South Africa [30]. Furthermore, the study identified a predominant Black African ethnicity (88.1%), with Coloured at 5.8% and Caucasian at 5.4%. This ethnic distribution reflects the demographics of the Eastern Cape Province, where the major-ity of the population is Black African [17]. Post-menopausal women constituted over three-quarters of the cohort (77.4%), a finding that contrasts with reports from other regions in Africa, such as Cameroon (58.6%) and Egypt (57%), where the majority of breast cancer patients were pre-menopausal [31,32]. This can be explained by the different molecular subtype

**Table 6.** Association of patients' breast cancer molecular subtypes with clinical characteristics.

| Variables | Luminal A | Luminal B | HER2 Enriched | TNBC | Total | P-value* |
|---|---|---|---|---|---|---|
| **N** | 78 (28.5%) | 130 (47.4%) | 15 (5.5%) | 51 (18.6%) | 274 (100.0%) | |
| **Breast Lump** | | | | | | |
| No | 1 (1.3%) | 1 (0.8%) | 0 (0.0%) | 2 (3.9%) | 4 (1.5%) | **0.458** |
| Yes | 77 (98.7%) | 129 (99.2%) | 15 (100.0%) | 49 (96.1%) | 270 (98.5%) | |
| **Breast lump + skin changes** | | | | | | |
| No | 47 (61.0%) | 61 (48.4%) | 7 (46.7%) | 34 (66.7%) | 149 (55.4%) | **0.087** |
| Yes | 30 (39.0%) | 65 (51.6%) | 8 (53.3%) | 17 (33.3%) | 120 (44.6%) | |
| **Breast lump + mastalgia** | | | | | | |
| No | 58 (77.3%) | 76 (61.3%) | 10 (66.7%) | 30 (61.2%) | 174 (66.2%) | **0.099** |
| Yes | 17 (22.7%) | 48 (38.7%) | 5 (33.3%) | 19 (38.8%) | 89 (33.8%) | |
| **Duration of presenting complaint** | | | | | | |
| <3 Months | 22 (28.6%) | 43 (34.4%) | 9 (60.0%) | 32 (62.7%) | 106 (39.6%) | **<0.001** |
| 3-6 Months | 21 (27.3%) | 48 (38.4%) | 5 (33.3%) | 15 (29.4%) | 89 (33.2%) | |
| >6 Months | 34 (44.2%) | 34 (27.2%) | 1 (6.7%) | 4 (7.8%) | 73 (27.2%) | |
| **Tumour size** | | | | | | |
| <2 cm | 8 (10.3%) | 11 (8.5%) | 3 (20.0%) | 1 (2.0%) | 23 (8.4%) | **0.120** |
| 2-5 cm | 31 (39.7%) | 37 (28.5%) | 5 (33.3%) | 16 (31.4%) | 89 (32.5%) | |
| >5 cm | 39 (50.0%) | 82 (63.1%) | 7 (46.7%) | 34 (66.7%) | 162 (59.1%) | |
| **Tumour size (cm)** | 5.7 (3.8) | 6.5 (3.5) | 6.7 (4.7) | 6.8 (3.3) | 6.4 (3.7) | **0.341** |
| **Clinical T stage** | | | | | | |
| T1 | 7 (9.0%) | 10 (7.7%) | 3 (20.0%) | 1 (2.0%) | 21 (7.7%) | **0.033** |
| T2 | 26 (33.3%) | 25 (19.2%) | 4 (26.7%) | 14 (27.5%) | 69 (25.2%) | |
| T3 | 11 (14.1%) | 20 (15.4%) | 1 (6.7%) | 15 (29.4%) | 47 (17.2%) | |
| T4 | 34 (43.6%) | 75 (57.7%) | 7 (46.7%) | 21 (41.2%) | 137 (50.0%) | |
| **Clinical N stage** | | | | | | |
| N0 | 45 (57.7%) | 54 (41.5%) | 5 (33.3%) | 31 (60.8%) | 135 (49.3%) | **0.118** |
| N1 | 22 (28.2%) | 36 (27.7%) | 4 (26.7%) | 11 (21.6%) | 73 (26.6%) | |
| N2 | 9 (11.5%) | 29 (22.3%) | 5 (33.3%) | 7 (13.7%) | 50 (18.2%) | |
| N3 | 2 (2.6%) | 11 (8.5%) | 1 (6.7%) | 2 (3.9%) | 16 (5.8%) | |
| **Clinical nodal status** | | | | | | |
| Negative | 45 (57.7%) | 54 (41.5%) | 5 (33.3%) | 31 (60.8%) | 135 (49.3%) | **0.023** |
| Positive | 33 (42.3%) | 76 (58.5%) | 10 (66.7%) | 20 (39.2%) | 139 (50.7%) | |
| **Clinical M stage** | | | | | | |
| M0 | 67 (85.9%) | 115 (88.5%) | 12 (80.0%) | 46 (92.0%) | 240 (87.9%) | **0.549** |
| M1 | 11 (14.1%) | 15 (11.5%) | 3 (20.0%) | 4 (8.0%) | 33 (12.1%) | |
| **Clinical TNM stage** | | | | | | |
| Stage Ia, Ib | 6 (7.7%) | 10 (7.7%) | 2 (13.3%) | 1 (2.0%) | 19 (6.9%) | **0.002** |
| Stage IIa, IIb | 33 (42.3%) | 28 (21.5%) | 4 (26.7%) | 26 (51.0%) | 91 (33.2%) | |
| Stage IIIa, IIIb, IIIc | 28 (35.9%) | 77 (59.2%) | 6 (40.0%) | 20 (39.2%) | 131 (47.8%) | |
| Stage IV | 11 (14.1%) | 15 (11.5%) | 3 (20.0%) | 4 (7.8%) | 33 (12.0%) | |
| **Early vs late BC** | | | | | | |
| Stage Ia, Ib, IIa | 26 (33.3%) | 22 (16.9%) | 5 (33.3%) | 13 (25.5%) | 66 (24.1%) | **0.037** |
| Stage IIb, III, IV | 52 (66.7%) | 108 (83.1%) | 10 (66.7%) | 38 (74.5%) | 208 (75.9%) | |

HER2, human epidermal growth factor receptor 2; TNBC, triple negative breast cancer; T, tumour; N, node; M, metastasis; BC, breast cancer

*$p$ values were calculated using the Fisher exact test

**Table 7. Association of the pathological characteristics and surgical intervention with the breast cancer molecular subtypes.**

| Variables | Luminal A | Luminal B | HER2 Enriched | TNBC | Total | P-value* |
|---|---|---|---|---|---|---|
| N | 78 (28.5%) | 130 (47.4%) | 15 (5.5%) | 51 (18.6%) | 274 (100.0%) | |
| **Tumour Histology** | | | | | | |
| Invasive ductal Ca (NOS) | 57 (74.0%) | 119 (92.2%) | 15 (100.0%) | 46 (90.2%) | 237 (87.1%) | **0.006** |
| Invasive lobular Ca | 6 (7.8%) | 6 (4.7%) | 0 (0.0%) | 2 (3.9%) | 14 (5.1%) | |
| Medullary Ca | 0 (0.0%) | 0 (0.0%) | 0 (0.0%) | 1 (2.0%) | 1 (0.4%) | |
| Mucinous Ca | 8 (10.4%) | 1 (0.8%) | 0 (0.0%) | 0 (0.0%) | 9 (3.3%) | |
| Papillary Ca | 6 (7.8%) | 2 (1.6%) | 0 (0.0%) | 1 (2.0%) | 9 (3.3%) | |
| Other | 0 (0.0%) | 1 (0.8%) | 0 (0.0%) | 1 (2.0%) | 2 (0.7%) | |
| **Histological grade** | | | | | | |
| Grade 1 | 26 (40.0%) | 6 (5.7%) | 0 (0.0%) | 0 (0.0%) | 32 (14.0%) | **<0.001** |
| Grade 2 | 36 (55.4%) | 75 (70.8%) | 5 (38.5%) | 12 (27.3%) | 128 (56.1%) | |
| Grade 3 | 3 (4.6%) | 25 (23.6%) | 8 (61.5%) | 32 (72.7%) | 68 (29.8%) | |
| **Surgery** | | | | | | |
| BCS | 0 (0.0%) | 4 (3.2%) | 0 (0.0%) | 2 (4.2%) | 6 (2.3%) | **0.037** |
| Mastectomy | 53 (68.8%) | 60 (48.0%) | 7 (46.7%) | 30 (62.5%) | 150 (56.6%) | |
| Nil | 24 (31.2%) | 61 (48.8%) | 8 (53.3%) | 16 (33.3%) | 109 (41.1%) | |

HER2, human epidermal growth factor receptor 2; TNBC, triple negative breast cancer; T, tumour; N, node; M, metastasis; BC, breast cancer; Ca, cancer; NOS, not otherwise specified; BCS, breast conserving surgery

*p values were calculated using the Fisher exact test

found in these regions, showing a triple negative breast cancer (TNBC) and human epidermal growth factor receptor 2 (HER2) enriched predominance compared to the Luminal B subtype predominance found in the current study. The TNBC and HER2 enriched molecular subtypes are more aggressive cancers and present at a younger age [33,34].

The most frequent presenting complaint reported was a breast lump (98.6%), which was either isolated or accompanied by symptoms such as skin changes (44.7%) or mastalgia (33.7%). This finding is consistent with studies from across SSA countries, such as Nigeria and Ethiopia [35,36]. A concerning observation from the study is the delayed presentation for care, with 61% of patients reporting a delay of over three months. Similar delays have been documented in other studies across SSA, including a study from Cameroon, which reported an average delay of 10.26 months from symptom onset to consultation [31], and a seven-month delay noted in Egypt [32]. Factors contributing to delayed presentation among patients in this study closely align with those reported in other African countries. Key issues identified include advanced age (over 40 years), low literacy rates, rural residency, insufficient knowledge about BC, lack of self-examinations and clinical evaluations, reliance on traditional medicine, and inadequacies in healthcare service availability [2].

In terms of tumour characteristics, the mean tumour size at diagnosis was 6.37 cm (±3.63), with over half (59.4%) of tumours exceeding 5 cm. In addition, the majority (50.4%) were diagnosed as advanced clinical T stage (T4) lesions. The tumour size aligns with variations seen across South Africa, as a study conducted at Potchefstroom Regional Hospital in the Northwest Province of South Africa revealed smaller tumours, with 49.3% measuring between 2 and 5 cm at diagnosis, but 61.5% of lesions being T4 lesions [30]. When examining tumour size across the broader context of SSA, northern African countries typically report earlier detection with smaller tumours; for instance, an average size of 2.8 cm in Algeria and 2.5 cm in Tunisia [37], compared to the larger tumours often seen in central African countries, as in the current study [7,10,36,38–40].

Lymph node involvement is crucial for long-term prognosis in BC [13]. In this study, about half (50.4%) of the women had palpable lymph nodes at diagnosis. This finding aligns with that of other studies in SSA, which also found advanced lymph node status at diagnosis [21,41,42].

The stage at which a patient presents bears a strong correlation with recurrence rates and survival outcomes in BC [43,44]. In this cohort, most patients were diagnosed at advanced Stages III and IV, with 60.1% of women presenting with advanced-stage disease (clinical Stage III in 48.2% of cases, and Stage IV in 11.9% of cases). Moreover, 11.9% of patients had developed metastasis by the time of presentation. Over three-quarters of patients (76.3%) presented at late stages (IIb, III, and IV). These findings correlate with trends observed in many SSA countries and highlight a critical need for improved awareness and early detection efforts. In contrast to Northern African nations, such as Morocco, which reports lower percentages of advanced-stage presentation (33%) [10,45], countries like Ethiopia (71.2%) and Nigeria (87.8%) show significantly higher advanced-stage cases at diagnosis [25,35]. This contrasts with HIC like the United States of America, where advanced-stage presentations are reported to be exceedingly rare (5–8%) [39,46].

The dominant histological type identified in this study was invasive ductal carcinoma not otherwise specified (NOS) (86.2%), with invasive lobular carcinoma (5.1%), papillary carcinoma (3.6%), and mucinous carcinoma (3.3%) being less prevalent. This predominance is corroborated by findings from other South African studies [27,41] and SSA studies [8,12,25]. Notably, the majority of tumours were classified as higher histological grades, specifically Grade 2 (56.2% of cases) and Grade 3 (29.8% of cases), indicating a trend towards aggressive phenotypes that are common among African women diagnosed with BC [21,27,31,41,42,47].

Surgical treatment was prevalent among participants, with 56.7% undergoing surgical procedures, mainly modified radical mastectomies (95.5%) and the remaining seven patients (4.5%) receiving breast conserving surgery (BCS). The reasons for the low rate of BCS in the study setting is multifactorial, there is limited access to specialized care as only one hospital in the area has the expertise to perform BCS, due to socioeconomic inequalities there is a lack of financial support to cover costs of long distance travel, accommodation and potentially missing work can hinder access to specialised care such as BCS and the fact that the mean clinical tumour size was 6.37 cm (SD ± 3.63), tumour size being >5 cm in 165 patients (59.4%), the most common clinical T stage being advanced T4 lesions (50.4%), half of the women had palpable axillary lymph nodes at diagnosis (n = 140; 50.4%) and 60.1% of the patients presented in advanced stage of disease (Stage III/IV) further hinders the rates of BCS. This aligns with the wider context of surgical interventions in SSA, where mastectomies tend to outnumber breast-conserving therapies owing to late presentations, which preclude patients from breast-conserving therapies, and resource limitations [15,25]. The reasons for the remaining 41.6% of patients not undergoing surgery in this study was due to 76.3% of the patients presented in late stage of disease, patient refusal due to cultural beliefs, lost to follow up and elderly patients not able to tolerate surgery.

The study also distinguished various molecular subtypes of BC, which provides insight into clinical behaviour, treatment response, and prognostic implications. The predominant molecular subtype identified was Luminal B (HER2–), accounting for 37.2% of cases. Following this were Luminal A (28.5%), (TNBC) (18.6%), Luminal B (HER2+) (10.2%), and HER2 enriched (5.5%) BC. This distribution highlights the diversity of BC characteristics based on molecular profiling, which is essential for determining individualised treatment strategies [27]. The findings indicate a marked variation in molecular subtype prevalence between HIC and LMIC nations, as well as within countries in the LMIC bracket. For instance, African populations exhibit a higher proportion of poor prognostic tumours, with TNBC and HER2 expression rates exceeding 20%, while hormone-positive BC rates are closer to 50%, compared to over 70% in high-income settings [8,48]. This study's findings align with those from similar studies conducted in South Africa, which note that over 75% of BC's are hormone receptor-positive, with Luminal B being the most common subtype [28]. However, differences emerged, as other South African studies indicated variability in subtype frequencies; Ooko et al. (2023) found Luminal A to be the predominant subtype (54%) in Limpopo Province, while Kakudji et al. (2020) reported Luminal A (29.5%) to be the most common subtype in Northwest Province [27,30].

The variation in molecular subtypes across African countries points to significant demographic and genetic influences on BC's incidence and presentation. For example, this study shows similar results to Egyptian studies indicate a higher frequency of luminal subtypes (up to 63.42%), which are associated with better prognoses and lower metastasis risks

[32,49]. In West Africa, the HER2 enriched subtype exhibited higher frequencies (14.87% in some regions) – almost triple that observed in this study (5.5%). Pooled data reveals that TNBC frequency hovers around 28.10% across the continent, with countries such as Ghana, Mali, and Nigeria reporting higher incidences at 56.17%, 47.85% and 40.32%, respectively [49]. These reported figures are more than double the percentage of TNBC found in the current study (18.6%). The findings suggest that factors such as ethnicity and genetics might play a crucial role in the expression of oestrogen and progesterone receptors among populations. African and African American women are five times less likely to express these receptors than other ethnicities, which can influence both prevalence and outcomes of BC [8,50].

The study also found significant associations between the duration of presenting complaints and molecular subtypes. HER2 enriched and TNBC subtypes tended to present earlier than the Luminal types, with Fisher's exact p-values significant at <0.001. TNBC and HER2 enriched subtypes are typically more aggressive molecular subtypes with a poorer prognosis [8,27]. Their aggressive growth could explain their earlier presentation. Given the prevalent poor prognoses associated with these subtypes [10,51], the study recommends robust BC screening initiatives to enhance early diagnosis in the Eastern Cape. Moreover, substantial relationships were observed between clinical T stage and molecular subtypes. The Luminal B subtype presented with the most advanced tumours (57.7% were T4). This pattern is similar to that found by Rayne et al. (2019) [14], suggesting a critical link between molecular characteristics and tumour advancement at diagnosis.

Lymph node involvement, indicating tumour infiltration beyond the primary disease, remains one of the foremost prognostic indicators in BC [13,27,42], and in this cohort, a significant correlation was identified between axillary involvement and specific molecular subtypes. The majority of HER2 enriched, and Luminal B cases exhibited axillary involvement (66.7% and 58.5%, respectively), while 42.3% of Luminal A and 39.2% of TNBC cases were similarly affected (p = 0.023). The presence of an axillary lump is essential in staging and has been linked to poorer outcomes [13], highlighting the importance of early detection and intervention. Previous research presented mixed results in the association between molecular subtypes and axillary lymph node status, with no association found in some cases [27,30]; however, findings in this study align with other literature that confirms significant relationships [52].

Furthermore, the current study established a notable connection between clinical stage at presentation and molecular subtypes, with a considerable proportion of Luminal B (70.7%) and HER2 enriched cases (60%) diagnosed at more advanced stages (Stages III or IV). Luminal B and TNBC subtypes had the highest proportion of patients presenting at a late stage (83.1% and 74.55, respectively). Previous studies from South Africa have yielded varying conclusions about the relationship between clinical staging and molecular subtypes. Ooko et al. (2023) suggested there was no association [27]; however, Joffe et al. (2018) indicated that advanced presentations were linked to Luminal B and TNBC subtypes [53], while a Nigerian study indicated that advanced disease was associated with TNBC and HER2 enriched subtypes, as in this study [54]. These results contrast with HIC like the United States of America (USA), where the Luminal BC presented mostly in Stage I and Stage II (80.4% and 68.1%, respectively) [55,56].

Histological analysis revealed a significant association between tumour type and molecular subtypes, with invasive ductal carcinoma NOS being the most common histological type across all categories: HER2 enriched (100%), Luminal B (92.2%), TNBC (90.2%), and Luminal A (74%). Notably, mucinous and papillary types were predominantly found in Luminal A cases, while invasive lobular carcinoma presented most frequently in both Luminal A (42.86%) and B (42.86%) subtypes. These findings are in alignment with a study conducted by Fatma (2019), which also noted the predominance of invasive ductal carcinoma NOS across various subtypes, the study also found that invasive lobular carcinoma was most common in Luminal A (86.9%) and Luminal B (11.9%) molecular subtypes [57], which is similar to the results in this study. The histological grading of tumours was closely related to molecular subtypes. In the current study, Grade 3 tumours were recorded in a substantial percentage of TNBC (72.7%) and HER2 enriched (61.5%) cases, while the majority of Luminal B tumours were categorised as Grade 2 (70.8%) and (40%) of Grade 1 tumours were found in the Luminal A subtype group. This is consistent with studies indicating that aggressive phenotypes, notably TNBC and HER2 enriched tumours,

frequently correlate with higher histological grades, while the Luminal A and B subtypes tend more towards lower grades [8,27,30].

Surgical interventions also varied by molecular subtype. The majority of patients with Luminal A and TNBC molecular subtypes underwent mastectomies (68.8% and 62.5%, respectively). These results echo findings of a prior study indicating a high association of surgical intervention with the TNBC subtype and advanced disease presentations [58]. The reason for surgical intervention in Luminal A molecular subtype is probably attributable to the less aggressive nature of Luminal BC, which often presents at more manageable stages, thus allowing for surgical intervention to be a viable option [27].

## Strength and limitations

This study stands out as the first to thoroughly investigate the clinicopathological characteristics and molecular subtypes of BC in the Eastern Cape, a resource-constrained and understudied province in South Africa. However, there are several limitations to note in this study. Firstly, some demographic and clinical data were missing owing to the retrospective nature of the study. Secondly, for analytical purposes the Luminal B (HER2- and HER2+) subtype group had to be combined to obtain greater numbers to allow for statistical analysis using the Fisher exact test. In addition, the study did not differentiate between triple-negative breast cancer and basal-like tumours, even though it is generally accepted that the triple-negative phenotype can serve as a proxy for basal-like characteristics. Furthermore, this is a single-centre study based in one geographic location and predominantly the Black African population; as such, other racial groups were underrepresented. Consequently, future studies spanning multiple centres and diverse population groups are recommended to enrich understanding of the epidemiological profile of BC in South Africa. This study did not assume a selection bias as it was limited by the data found in the medical records.

## Conclusions

This study highlights critical insights into BC epidemiology in the Eastern Cape, South Africa, revealing late-stage presentation, larger tumours, and delays in seeking care. The predominance of high-grade and aggressive molecular subtypes underscores the need for effective screening and awareness programmes. Future research should determine the survival rates and the effectiveness of the national breast cancer screening programme. Strengthening regional health policies to improve screening access and early detection is essential. Public health strategies, including education and timely referrals, are vital for reducing breast cancer-related morbidity and mortality in the area.

## Supporting information

**S1 Table. Anonymized recorded data.**
(XLSX)

## Acknowledgments

We acknowledge E.J.K. Simpson and the Department of General Surgery, Frere Hospital, for their support throughout the research process.

## Author contributions

**Conceptualization:** Hayden Gerald Kretzmann.

**Data curation:** Hayden Gerald Kretzmann.

**Formal analysis:** Oladele Vincent Adeniyi.

**Methodology:** Hayden Gerald Kretzmann.

**Resources:** Oladele Vincent Adeniyi.

**Supervision:** Oladele Vincent Adeniyi.

**Visualization:** Oladele Vincent Adeniyi.

**Writing – original draft:** Hayden Gerald Kretzmann.

**Writing – review & editing:** Hayden Gerald Kretzmann, Oladele Vincent Adeniyi.

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
