## [Decision Letter · Decision Letter 0]

PONE-D-25-13977The Clinicopathological and Molecular Profile of Breast Cancer in a Tertiary Hospital in the Eastern Cape Province, South Africa.PLOS ONE

Dear Dr. Kretzmann,

Thank you for submitting your manuscript to PLOS ONE. After careful consideration, we feel that it has merit but does not fully meet PLOS ONE’s publication criteria as it currently stands. Therefore, we invite you to submit a revised version of the manuscript that addresses the points raised during the review process.

Please revise the articl in the light of comments of the reviewers, given at the end of this message. 

We look forward to receiving your revised manuscript.

Kind regards,

Abdul Rauf Shakoori, PhD

Academic Editor

PLOS ONE

Journal Requirements:

3. Please amend the manuscript submission data (via Edit Submission) to include author Dr. Oladele Vincent Adeniyi.

Reviewers' comments:

Reviewer's Responses to Questions

**Comments to the Author**

1. Is the manuscript technically sound, and do the data support the conclusions?

Reviewer #1: Partly

Reviewer #2: Yes

2. Has the statistical analysis been performed appropriately and rigorously? 

Reviewer #1: Yes

Reviewer #2: Yes

3. Have the authors made all data underlying the findings in their manuscript fully available?

Reviewer #1: No

Reviewer #2: Yes

4. Is the manuscript presented in an intelligible fashion and written in standard English?

Reviewer #1: Yes

Reviewer #2: Yes

5. Review Comments to the Author

Reviewer #1: This is a well-conducted, timely, and comprehensive retrospective analysis highlighting the clinicopathological and molecular profile of breast cancer patients from an underrepresented region in South Africa. The study fills an important data gap and presents detailed insights that are regionally and globally relevant, especially for low- and middle-income country (LMIC) settings. The manuscript is overall well-structured and thorough. However, there are areas that would benefit from clarification, deeper discussion, and minor editing for consistency and precision.

The molecular classification based on IHC is appropriate, but combining Luminal B (HER2– and HER2+) may oversimplify biologically distinct entities. Since numbers for HER2+ tumors are small, this is understandable for analysis, but it should be more explicitly discussed as a limitation.

A table summarizing significant associations (e.g., subtype vs grade, stage, delay, etc.) in a concise multivariate format would greatly enhance readability.

The finding that only 4.5% underwent breast conservation surgery is critical. The authors should comment more explicitly on the infrastructure, surgical expertise, and patient-related factors that influence this low rate.

The data on non-surgical management (41.6% did not undergo surgery) needs elaboration—was this due to stage IV disease, resource limitation, or patient refusal?

Minor Comments

Title: Consider revising to highlight the focus on molecular subtypes:

“Clinicopathological and Molecular Subtypes of Breast Cancer in Eastern Cape, South Africa: A Two-Year Retrospective Study”.

Abstract: Very informative. However, avoid redundancy (e.g., phrase “BC molecular subtypes of BC”).

Tables: Tables are well-prepared but very dense. Some tables can be moved to supplementary material if needed.

Grammar and Style:

Some repetition (e.g., “clinicopathological and molecular profile of BC”) should be avoided.

Ensure consistency in use of acronyms (e.g., always define TNBC, HER2 at first mention in each section).

Reviewer #2: Dear authors, thank you for this interesting manuscript, here are my comments for more clarity:

#Missing Data Handling: The limitations section notes missing demographic and clinical data due to the retrospective nature, but methods do not specify how this was addressed (e.g., exclusion, imputation). Clarification is essential for transparency.

#Consider including STROBE checklist as supporting information for transparency.

# Standardize table formatting (e.g., alignment, decimal places, abbreviations like "NOS" and "TNBC" should be defined in footnotes).

#Correct minor errors (e.g., "timeous referral" → "timely referral"; "HER2-enriched in 18.6% and 5.5% of cases" → "HER2-enriched in 5.5% of cases").

# Ensure consistent terminology (e.g., "LMIC" vs. "LMC"; define all abbreviations at first mention).

6. PLOS authors have the option to publish the peer review history of their article (what does this mean? ). If published, this will include your full peer review and any attached files.

**Do you want your identity to be public for this peer review?** For information about this choice, including consent withdrawal, please see our Privacy Policy .

Reviewer #1: **Yes: ** Sanjay Kumar Yadav

Reviewer #2: No

---

## [Author Response · Author response to Decision Letter 1]

6 May 2025

Thank you for the comments and suggestions made.

We have addressed all comments and suggestions in the Track Changes Manuscript and have responded with a rebuttal letter addressing all concerns raised by the editors.

Thank you again for all the suggestions.

I have also made the adjustments to the title, anonymized the data and moved the ethics statement to the methods section in the revised manuscript with track changes.

---

## [Editor Report · Decision Letter 1]

Clinicopathological and Molecular Subtypes of Breast Cancer in the Eastern Cape, South Africa: A Two-Year Retrospective Study

PONE-D-25-13977R1

Dear Dr. Kretzmann,

We’re pleased to inform you that your manuscript has been judged scientifically suitable for publication and will be formally accepted for publication once it meets all outstanding technical requirements.

Kind regards,

Abdul Rauf Shakoori, PhD

Academic Editor

PLOS ONE

Additional Editor Comments (optional):

NIL
---

## [Editor Report · Acceptance letter]

PONE-D-25-13977R1

PLOS ONE

Dear Dr. Kretzmann,

I'm pleased to inform you that your manuscript has been deemed suitable for publication in PLOS ONE. Congratulations! Your manuscript is now being handed over to our production team.

Kind regards,

on behalf of

Prof. Dr. Abdul Rauf Shakoori

Academic Editor

PLOS ONE